# Incidence, demographics, characteristics and management of acute Achilles tendon rupture: An epidemiological study

**Samuel Briggs-Price**[1]*, **Jitendra Mangwani**[2], **Linzy Houchen-Wolloff**[3], **Gayatri Modha**[4], **Emma Fitzpatrick**[2], **Murtaza Faizi**[2], **Jenna Shepherd**[2], **Seth O'Neill**[1]

1 School of Healthcare, University of Leicester, Leicester, United Kingdom, 2 Orthopaedics, University Hospitals of Leicester, Leicester, United Kingdom, 3 Centre for Exercise and Rehabilitation Sciences, NIHR Biomedical Research Centre, Leicester, United Kingdom, 4 Emergency Care, University Hospitals of Leicester, Leicester, United Kingdom

* sbp18@leicester.ac.uk

## Abstract

### Background

Achilles tendon rupture (ATR) account for 10.7% of all tendon and ligament injuries and causes lasting muscular deficits and have a profound impact on patients' quality of life. The incidence, characteristics and management of ATR in the United Kingdom (UK) is poorly understood. This investigation aims to understand the incidence of ATR in the UK.

### Methods

Prospective data collection of ATR incidence from a United Kingdom Emergency department. Retrospective review of management protocols and immobilisation duration from electronic medical records.

### Results

ATR incidence is 8 per 100,000 people per annum. Participants were predominately male (79.2%) and primarily reported a sporting mechanism of injury (65.2%). Mean immobilisation duration was 63.1 days. 97.1% were non-surgically managed post ATR. 46.2% of participants had experienced a previous ATR or Achilles tendinopathy prior to their current ATR.

### Conclusion

The incidence of ATR found was 8. cases per 100,000 people per annum. Most ATR were managed non-surgically in this cohort. The majority of ruptures occurred during sporting activity. Almost one quarter (23.3%) of individuals report Achilles pain prior to ATR.

**Data Availability Statement:** All relevant data are available at figshare (https://doi.org/10.25392/leicester.data.24948018).

**Funding:** The author(s) received no specific funding for this work.

**Competing interests:** The authors have declared that no competing interests exist.

**Abbreviations:** AT, Achilles Tendon; ATR, Achilles Tendon Rupture; UK, United Kingdom; LAMP, Leicester Achilles Management Protocol; SMART, Swansea Morriston Achilles Rupture Treatment; ED, Emergency Department; NHS, National Health Service; SD-, Standard Deviation; IQR, Interquartile Range.

## Background

The Achilles tendon (AT) is the largest and strongest tendon in the body [1]. Through elastic energy storage, the AT improves movement efficiency, transmitting forces of 2.7–3.95 times body weight during walking and 4.15 to 7.71 when running [2]. Achilles tendon ruptures (ATR) can occur when tendon strain exceeds maximum tendon capacity. Common mechanisms for ATR include sudden or violent dorsiflexion of the ankle or a sporting acceleration-deceleration mechanism [3]. ATR are the most common tendon ruptures accounting for 10.7% of all tendon and ligament injuries [4]. Incidence rates range is 2.5–47 per 100,000 person-years in north America and Europe [5–10]. ATR incidence is rising, with the most significant increase between the ages of 40–59 [6, 7]. Significant variation in incidence occurs due to the population sampled (male/female), sample age, geographic range (local/regional/national), sampling setting (emergency department/medical database review) or time of sampling (season). UK ATR incidence increased from 6–13 per 100,00 person years from 1995 to 2019 [11, 12]. However, this data represents ATR incidence in Scotland and a single NHS trust in England. Further incidence data is required to improve understanding of ATR incidence in England.

Risk factors for ATR can be categorised as intrinsic and extrinsic. Intrinsic factors include AT properties, age, sex, genetics and systemic comorbidities. Extrinsic factors include sporting activity, exposure to AT loading and medications [13, 14]. Previous studies have proposed that the rising incidence of ATR is associated with the increasing use of medications associated with ATR in an active, aging population [15]. However, the quality of evidence remains low and further research is required to understand the demographics of the ATR population in the UK.

Surgical and non-surgical management approaches following ATR have been compared extensively. In the UK, non-surgical approaches represent standard practice due to comparable rates of re-rupture with accelerated functional rehabilitation protocols and lower associated complications [16–20]. Recent randomised controlled trials have reported an increased re-rupture risk following non-surgical management [21]. However, in this trial, non-surgical management did not represent the typical accelerated functional rehabilitation non-surgical management protocols that have been developed in the UK, such as the Leicester Achilles Management Protocol (LAMP) and Swansea Morriston Achilles Rupture Treatment (SMART) protocol [22, 23]. AT re-ruptures are reported between 0.9–2% when adopting these non-surgical protocols. However, due to significant loss to follow up, the studies investigating these protocols may under-represent non-surgical re-rupture rates and should be interpreted with caution.

This investigation aims to 1) identify the incidence of ATR in the UK 2) identify the characteristics of the ATR population 3) Identify rates of surgical and non-surgical management including duration of boot immobilisation.

## Methods

### Trial design

Retrospective analysis of prospectively collected data of individuals presenting to the emergency department (ED) diagnosed with ATR. Data was collected at a National Health Service (NHS) Trust from March 2015 to June 2021.

### Participants

All individuals with clinically confirmed ATR documented in ED medical notes were included in the analysis. Comorbidity and medication data was extracted from ED medical notes. In

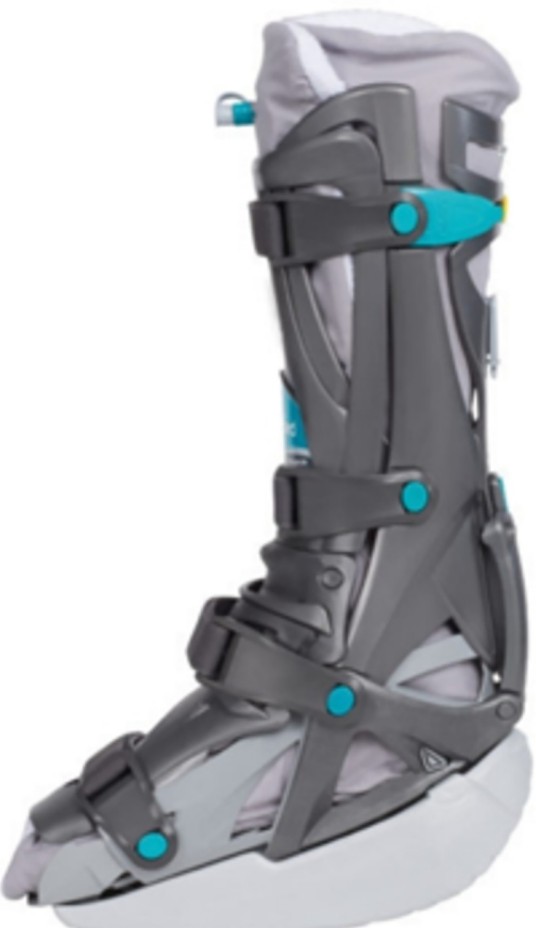

**Fig 1. Vacoped controlled mobilisation boot.**

addition to routine comorbidity data collection, participants were asked if they had experienced previous Achilles injury or pain. Medical records were reviewed retrospectively to determine management protocols (surgical/non-surgical) and controlled mobilisation (VACOped™ boot) (Fig 1) duration. The NHS trust studied routinely uses the Leicester Achilles Management Protocol (LAMP) [22] consisting of 8 weeks-controlled mobilisation.

## Data analysis

Data was analysed using SPSS (V28.0, IBM, New York, USA). Data distribution was assessed for normality and reported as means/ median with standard deviation (SD)/ interquartile range (IQR).

Annual incidence data was calculated using complete years. Population data was taken from the County Council demography report [24]. Comorbidity/ medication data used for comparison was taken from NHS or musculoskeletal health reports [25–27].

The relationship between mechanism of injury, demographics, comorbidities and medications are explored. The management approach (surgical/non-surgical) was analysed in relation to participant demographics, comorbidities and duration to present to ED. Groups were compared using an independent t-test or non-parametric equivalent.

**Table 1. Number and classification of comorbidities.**

|  | ATR Participants (n = 328) | General Population |
|---|---|---|
| Number of Comorbidities Mean (SD) | 1.2 (1.5) |  |
| Diabetes n (%) | 22 (6.9) | 6.5% |
| Hypertension n (%) | 40 (12.6) | 13.8% |
| Cardiovascular Disease n (%) | 33 (10.3) | 9.1% |
| Rheumatological condition n (%) | 3 (1.0) | 0.7% |
| Respiratory n (%) | 31 (9.8) | 7.8% |
| Other Musculoskeletal n (%) | 70 (22.1) | 32% |

## Results

There were 361 participants diagnosed with ATR in ED and included in analysis. Participant's median (IQR) age was 45 (19) years and were predominantly male (79.2%). Comorbidity and medication data was available for 328 and 316 participants. Comorbidities and medications documented on ED assessment and the comparison to local prevalence are displayed in Tables 1 and 2. Concurrent with routine data collection, pre-rupture Achilles pain/ injury data was collected from 117/361 participants. Achilles pain/ injury prior to ATR was reported by 46.2% (n = 54/117) of participants (Table 3). In participants with Achilles pain/ injury, 61.1% (n = 33/54) reported their complaint bilaterally or on the side of current ATR. There was no significant difference in the number of comorbidities between participants who reported previous AT pain or rupture and participants with no prior symptoms.

The primary mechanism of injury was sporting activities (65.2%). Participants with a sporting mechanism of injury had a mean (SD) age of 41.0 (12.2) years in comparison to 55.4 (13.4) in participants with a non-sporting mechanism of injury (p = 0.55). Participants with a sporting mechanism of injury had statistically significantly fewer comorbidities (0.9 (1.3) vs 1.7 (1.8), p = <0.001) and medications (0.7 vs 1.7, p = <0.001) than those with a non-sporting mechanism of injury.

The median (IQR) time to present to ED following injury was 0 days (1). Nearly all (97.1%) of the participants were non-surgically managed post ATR. There were no significant differences in age or number of comorbidities between surgical and non-surgical groups. The surgical group had a greater mean (days) duration between initial injury and presenting to ED (2.6 (4.9) vs 1.9 (4.8), p = 0.36). The mean (SD) duration wearing the immobilisation boot was 63.1 (10.8) days.

**Table 2. Number and classification of medications.**

|  | ATR Participants (n = 316) |  | General Population |
|---|---|---|---|
| Number of Medications Mean (SD) | 1.1 (1.8) |  |  |
| Statin n (%) | 37 (11.7) |  | 14% |
| Antihypertensive n (%) | 46 (14.6) |  | 15% |
| Analgesia n (%) | 14 (4.4) |  | 11% |
| Proton Pump Inhibitors n (%) | 18 (5.7) |  | 11% |
| Steroids n (%) | 29 (9.2) | Inhaled:17 (58.6) |  |
|  |  | Injected: 5 (17.2) |  |
|  |  | Oral:7 (24.1) |  |
| Antibiotics (fluoroquinolones) (%) | 2 (0.6) |  |  |
| Anticoagulants (%) | 5 (1.6) |  | 5% |
| Anti-inflammatories (%) | 13 (4.1) |  | 11% |

**Table 3. Achilles pain/injury pre-rupture.**

| Achilles Pain/ Injury | n (%) |
|---|---|
| Contralateral Achilles Rupture (n = 117) | 19 (16.2) |
| Ipsilateral Achilles Rupture (n = 117) | 7 (6.0) |
| Contralateral Achilles Tendinopathy (n = 112) | 2 (1.8) |
| Ipsilateral Achilles Tendinopathy (n = 112) | 16 (14.3) |
| Bilateral Achilles Tendinopathy (n = 112) | 10 (8.9) |
| Total participants reporting Achilles Pain/Injury Pre-Rupture (n = 117) | 54 (46.2) |

The ATR per month incidence rate for each year (March 2016-June 2021) is provided in Fig 2 for all participants.

Annual ATR incidence was calculated for complete years (2016–2020, n = 277). The mean (SD) annual incidence was 56 (6) ATR per year. An incidence rate of 8 per 100,000 people per annum.

The reference line shows the line of best fit over the last 6 years of ATR data.

## Discussion

The incidence of ATR presenting to ED was 8 per 100,000 people per annum. Consistent with previous UK ATR data, an increasing incidence of ATR was identified [11]. International ATR incidence rates are varied, these findings report a higher incidence than the United States, 3.2 per 100,000 person years [7] but lower than other nations (29.5–32.3 per 100,000 person years) [5, 6]. Incidence reports in the United States represent data from an electronic database review [7]. As all medical centres were not included, this review is unlikely to capture all ATR and is expected to under-represent the true incidence rate in the United States.

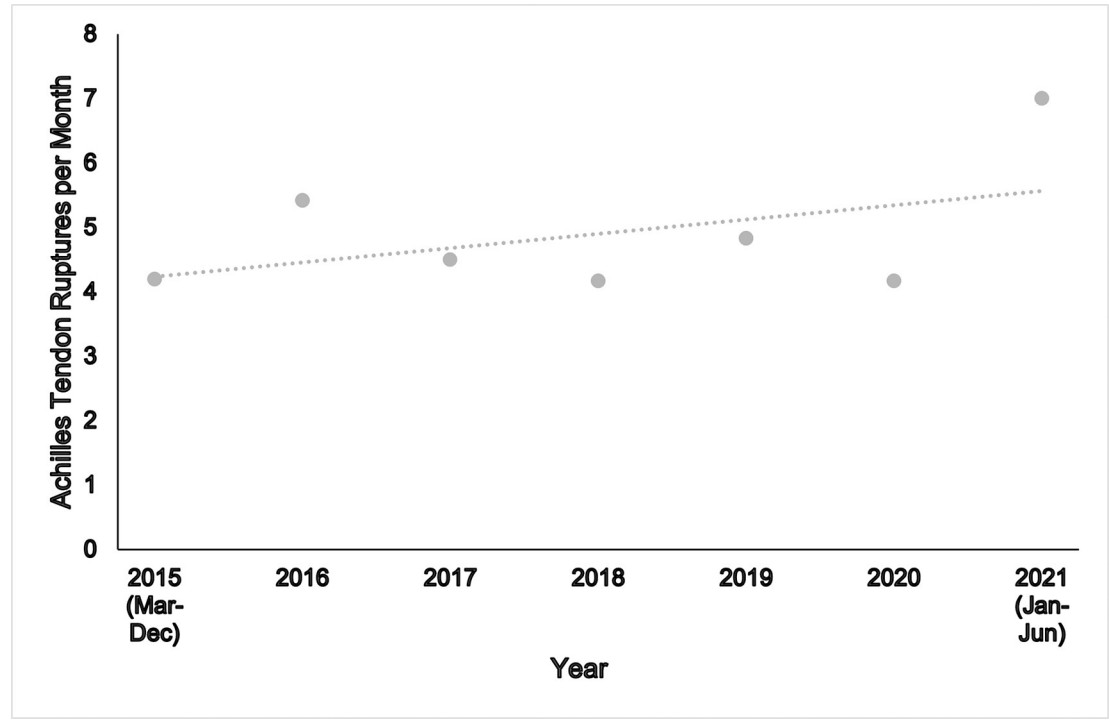

**Fig 2. Achilles tendon rupture incidence per month.**

The ATR population is predominantly male, and injuries most commonly occur during sporting activities. This is consistent with current incidence studies [5–7]. However, males represented a greater proportion of participants and ATR were more frequently associated with non-sporting mechanisms than previously observed [7, 28].

Medications and comorbidities may contribute to ATR aetiology [13, 14]. The overall distribution of comorbidities and medications were similar to the general population. Fluroquinolones were only reported in two cases despite the known association with ATR [29]. The medication dose and interaction between concomitant medications was not investigated and may have a significant impact on ATR risk. The lack of comorbidities and medications found may be due to ATR primarily occurring in the sporting population who present with less comorbid health conditions. In this study, the non-sporting population are older and have greater number of comorbidities and medications. Identifying an older subgroup who present with greater number of comorbidities may provide an insight into tailoring future immobilisation and rehabilitation protocols.

Previous ATR or pain was reported in nearly half of participants. This is the first UK data on Achilles symptoms prior to rupture. As data collection was additional to usual care the sample was limited and findings should be interpreted with caution. In comparison, previous studies found one third of participants experience Achilles symptoms prior to ATR [30]. Achilles tendinopathy is a risk factor for ATR with 4% of individuals with Achilles tendinopathy experiencing ATR [14, 30]. It is unknown if current interventions for AT pathologies reduce the risk of ATR as the ability to alter tendon structure is debated [31].

The UK management of ATR is predominantly non-surgical. This contrasts with surgical management rates seen internationally [5, 6]. The immobilisation duration is consistent with published protocols in this region of the UK [22]. There is a disparity with other regions in the UK which adopt longer immobilisation periods [23].

## Study limitations

This study was limited to a single ED site, it is expected that primary care sites across the region that were not included in this analysis will manage acute ATR. Therefore, the incidence reported is conservative and ATR presenting to secondary care is anticipated to be higher. Future research needs to include larger numbers of primary care sites across the UK to determine the true incidence rate and management of ATR. Understanding current ATR incidence and management is essential for the development of future care in this population.

## Conclusion

The incidence rate of ATR in England is higher than previously reported elsewhere in the UK. There is a continuing trend towards increasing ATR incidence each year. Non-sporting mechanisms of injury are more common than previously reported and occur in an older population with greater number of comorbidities and medications.

## Acknowledgments

Authors would like to thank the University Hospitals of Leicester NHS Trust healthcare professionals who have supported data collection.

## Author Contributions

**Conceptualization:** Samuel Briggs-Price, Jitendra Mangwani, Linzy Houchen-Wolloff, Seth O'Neill.

**Data curation:** Samuel Briggs-Price, Jitendra Mangwani, Gayatri Modha, Emma Fitzpatrick, Murtaza Faizi, Jenna Shepherd, Seth O'Neill.

**Formal analysis:** Samuel Briggs-Price, Seth O'Neill.

**Investigation:** Samuel Briggs-Price.

**Methodology:** Samuel Briggs-Price, Seth O'Neill.

**Supervision:** Linzy Houchen-Wolloff, Seth O'Neill.

**Writing – original draft:** Samuel Briggs-Price.

**Writing – review & editing:** Samuel Briggs-Price, Jitendra Mangwani, Linzy Houchen-Wolloff, Gayatri Modha, Emma Fitzpatrick, Murtaza Faizi, Jenna Shepherd, Seth O'Neill.

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
