## [Decision Letter · Decision Letter 0]

19 Dec 2023

PONE-D-23-38265Incidence, demographics, characteristics and management of acute Achilles tendon rupturePLOS ONE

Dear Dr. Briggs-Price,

Thank you for submitting your manuscript to PLOS ONE. After careful consideration, we feel that it has merit but does not fully meet PLOS ONE’s publication criteria as it currently stands. Therefore, we invite you to submit a revised version of the manuscript that addresses the points raised during the review process.

We look forward to receiving your revised manuscript.

Kind regards,

Charlie M. Waugh

Academic Editor

PLOS ONE

Journal Requirements:

2. In the online submission form, you indicated that The datasets generated and/or analysed during the current study are not publicly available due to University Hospitals of Leicester data management policy but are available from the corresponding author on reasonable request.

Reviewers' comments:

Reviewer's Responses to Questions

**Comments to the Author**

1. Is the manuscript technically sound, and do the data support the conclusions?

Reviewer #1: Yes

Reviewer #2: Yes

2. Has the statistical analysis been performed appropriately and rigorously? 

Reviewer #1: Yes

Reviewer #2: Yes

3. Have the authors made all data underlying the findings in their manuscript fully available?

Reviewer #1: No

Reviewer #2: Yes

4. Is the manuscript presented in an intelligible fashion and written in standard English?

Reviewer #1: Yes

Reviewer #2: Yes

5. Review Comments to the Author

Reviewer #1: Dear Authors,

the aim of your manuscript is to report epidemiological data about Achilles tendon ruptures in the UK.

The article is overall well-written and well-organized.

I have some suggestions to improve the quality of the manuscript:

1) The Title may be misleading: I would specifically report that this is an epidemiological study.

2) Introduction is short but fine. You may add something more on how an ATR can occur, focusing on what kind of specific movements (eg. accelerating after landing with violent ankle dorsiflexion) and comorbidities/drugs (eg. diabetes, fluoroquinolones, etc). This could be helpful for a better uderstantind of your results.

3) Methods: it should be reported "who made what".

4) Methods: a Figure of the VACOped boot and a biref description could be useful.

5) Results: Figure 1 is missing.

6) Discussion: I suggest to give short interpretation of your results, linking them to some specific observed conditions or drugs (eg. diabetes, higher statins consumption, sterioids, antibiotics, etc).

7) Please consider to cite the following articles:

- Tarantino D, Palermi S, Sirico F, Corrado B. Achilles Tendon Rupture: Mechanisms of Injury, Principles of Rehabilitation and Return to Play. J Funct Morphol Kinesiol. 2020 Dec 17;5(4):95. doi: 10.3390/jfmk5040095. PMID: 33467310; PMCID: PMC7804867.

- Tarantino, D., Palermi, S., Sirico, F., Balato, G., D‘Addona, A., & Corrado, B. (2020). Achilles tendon pathologies: How to choose the best treatment. Journal of Human Sport and Exercise, 15(4proc), S1300-S1321. doi:https://doi.org/10.14198/jhse.2020.15.Proc4.29

- Tarantino D, Mottola R, Resta G, Gnasso R, Palermi S, Corrado B, Sirico F, Ruosi C, Aicale R. Achilles Tendinopathy Pathogenesis and Management: A Narrative Review. Int J Environ Res Public Health. 2023 Aug 30;20(17):6681. doi: 10.3390/ijerph20176681. PMID: 37681821; PMCID: PMC10487940.

Reviewer #2: General comments

Important study of a topic with increasing interest. Well written manuscript. There are some issues that need to be addressed as explained below. The discrepancy of the incidence data between this study and the incidences reported in Finland and Denmark is a concern. The authors briefly discuss this in “Limitations”. Overall, only minor revision seems necessary before publication.

Specific comments

Abstract

Well written. But why do you not state “symptomatic” instead of “asymptomatic” in the last sentence in the conclusion? Most ATR occur in individuals with no prior pain before rupture…

Background

Line 7-8: Regarding incidences in north America and Europe. The study by Lemme et al (ref. 6) used data from the National Electronic Injury Surveillance System (NEISS) database which collects data from approximately 100 representative US hospitals with at least 6 beds and with a 24-hours emergency department services. Achilles tendon ruptures treated outside these kinds of hospitals (non-operatively or surgically) will not be registered in the database. Therefore, the data from the Lemme study is most probably not representative for the true incidence of acute Achilles tendon rupture in the US. The incidence is most probably higher. The observed increase in incidence, however, is probably realistic. Suggest to either comment on this or remove the reference and the incidence numbers referred to in the text.

Or consider using the article from Cretnik instead of Lemme and narrow the incidence interval according to the incidence in the remaining publications (Čretnik A, Košir R. Incidence of Achilles tendon rupture: 25-year regional analysis with a focus on bilateral ruptures. J Int Med Res. 2023 Nov;51(11):3000605231205179. doi: 10.1177/03000605231205179. PMID: 37976267; PMCID: PMC10657533.)

The authors might also consider using this reference: Ganestam A, Kallemose T, Troelsen A, Barfod KW. Increasing incidence of acute Achilles tendon rupture and a noticeable decline in surgical treatment from 1994 to 2013. A nationwide registry study of 33,160 patients. Knee Surg Sports Traumatol Arthrosc. 2016 Dec;24(12):3730-3737. doi: 10.1007/s00167-015-3544-5. Epub 2015 Feb 20. PMID: 25697284.

Line 18-21: The authors claim there is similar risk of re-rupture comparing surgical and non-surgical treatment. This is most probably an assumption based on RCTs wich are under-powered (ref. 10-13). The largest RCT performed on treatment results following ATR clearly finds a significant higher incidence of re-rupture after non-operative treatment (Myhrvold SB, Brouwer EF, Andresen TKM, Rydevik K, Amundsen M, Grün W, Butt F, Valberg M, Ulstein S, Hoelsbrekken SE. Nonoperative or Surgical Treatment of Acute Achilles' Tendon Rupture. N Engl J Med. 2022 Apr 14;386(15):1409-1420. doi: 10.1056/NEJMoa2108447. PMID: 35417636.)

I would also suggest the authors to read the editorial of Barfod and Hölmich stating that surgery protects from re-rupture (Barfod KW, Hölmich P. Acute Achilles' Tendon Rupture - Surgery or No Surgery. N Engl J Med. 2022 Apr 14;386(15):1465-1466. doi: 10.1056/NEJMe2202696. PMID: 35417642.)

Please change this section according to available high-quality publications and consider adding these publications to the reference list.

METHODS

Trial Design

OK.

RESULTS

Tables 1, 2 and 3:

Please remove the “%” from the parentheses in the second column.

DISCUSSION

Line 129: Please replace “prevalence of ATR” with “incidence of ATR”.

Line 130: Again (as in line 7-8): The low incidence data from US is not trustworthy for the true incidence of ATR. Suggest to either comment on this or rephrase.

Study Limitations:

Well written. The true incidence of ATR in the UK is most probably higher that the incidence reported here. The authors are obviously aware of that.

Referenses:

I have concerns of two of the references:

Firstly ref. 6 (Lemme et al). See my explanation above under Background.

Secondly ref. 15 (Hutchison). This is a study which should be interpreted with a high degree of caution. I suggest the authors to read the answer to the Editor in tbe NEJM from 2022 (Myhrvold SB, Ulstein S, Hoelsbrekken SE. Nonoperative or Surgical Treatment of Acute Achilles' Tendon Rupture. Reply. N Engl J Med. 2022 Jul 7;387(1):91. doi: 10.1056/NEJMc2206333. PMID: 35793214.) Please take this into consideration and adapt this knowledge of potential loss to follow-up in that study when referring to this throughout the manuscript. Or consider removing the reference.

Good luck with the revision!

6. PLOS authors have the option to publish the peer review history of their article (what does this mean?). If published, this will include your full peer review and any attached files.

Reviewer #1: No

Reviewer #2: No

---

## [Author Response · Author response to Decision Letter 0]

15 Jan 2024

Thank you for taking the time to review the manuscript. Full response to reviewer comments has been included in the attached files

---

## [Decision Letter · Decision Letter 1]

21 Feb 2024

PONE-D-23-38265R1Incidence, demographics, characteristics and management of acute Achilles tendon rupture: an epidemiological studyPLOS ONE

Dear Dr. Briggs-Price,

Thank you for submitting your manuscript to PLOS ONE. After careful consideration, we feel that it has merit but does not fully meet PLOS ONE’s publication criteria as it currently stands. Therefore, we invite you to submit a revised version of the manuscript that addresses the points raised during the review process.

We look forward to receiving your revised manuscript.

Kind regards,

Charlie M. Waugh

Academic Editor

PLOS ONE

Journal Requirements:

**Additional Editor Comments:**

One of the reviewers has additional concerns that warrant investigation. I am returning the manuscript in the event you wish to address this comment and amend the manuscript.

Reviewers' comments:

Reviewer's Responses to Questions

**Comments to the Author**

1. If the authors have adequately addressed your comments raised in a previous round of review and you feel that this manuscript is now acceptable for publication, you may indicate that here to bypass the “Comments to the Author” section, enter your conflict of interest statement in the “Confidential to Editor” section, and submit your "Accept" recommendation.

Reviewer #1: All comments have been addressed

Reviewer #2: (No Response)

2. Is the manuscript technically sound, and do the data support the conclusions?

Reviewer #1: Yes

Reviewer #2: Yes

3. Has the statistical analysis been performed appropriately and rigorously? 

Reviewer #1: Yes

Reviewer #2: I Don't Know

4. Have the authors made all data underlying the findings in their manuscript fully available?

Reviewer #1: Yes

Reviewer #2: No

5. Is the manuscript presented in an intelligible fashion and written in standard English?

Reviewer #1: Yes

Reviewer #2: Yes

6. Review Comments to the Author

Reviewer #1: (No Response)

Reviewer #2: Review of PONE-D-23-38265_R1

Incidence, demographics, characteristics and management of acute Achilles tendon rupture: an epidemiological study

Concerns

It seems like the authors have addressed most of the concerns and suggestions from the reviewers. The revision of the manuscript is an improvement. However, I would like to highlight three minor and one major concern of the revised manuscript:

1) Minor concern: The authors found that the incidence is 8 ATR per 100.000 people/year. This is stated both in the results and conclusion sections in the abstract and the manuscript. At the same time the authors discuss the possibility of this number to be an underestimation of the true incidence. I would suggest rephrasing these sentences and just state that you found an incidence of 8 ATR / 100.000 / year, not that it is.

2) Minor concern: Line 10-11: The authors claim that the incidence rates range “from 2.5-32.2”.... Suggest to instead use “range from NN to NN” or “the incidence range is NN-NN”. In addition, the authors have not considered the incidence data from Sweden published by Huttunen et al in 2014. They found an incidence of almost 47 ATR / 100000 / year in 2012. Please include this reference and adjust the incidence range accordingly. (Huttunen TT, Kannus P, Rolf C, Felländer-Tsai L, Mattila VM. Acute achilles tendon ruptures: incidence of injury and surgery in Sweden between 2001 and 2012. Am J Sports Med. 2014 Oct;42(10):2419-23. doi: 10.1177/0363546514540599. Epub 2014 Jul 23. PMID: 25056989.)

3) Minor concern: From author guidelines: “References are listed at the end of the manuscript and numbered in the order that they appear in the text.” Please comply with the author guidelines.

4) Major concern: Lines 28-37: Although most ATRs can be managed safely without surgery if treatment with equinus positioning of the ankle is initiated early, the authors still claim that the re-rupture rates after surgical and non-operative treatment are similar. The authors have performed some changes to this section and implemented reference to the RCT published in NEJM in 2022 (ref. 28). They state, however, that the non-surgical management did not represent protocols that have been developed in the UK such as the LAMP and SMART protocol. This is a major concern regarding bias. The fact that other protocols are developed outside the UK is scientifically irrelevant. The major adverse event with the highest negative impact on ATRS and return to sports has shown to be re-rupture (Metz R, van der Heijden GJ, Verleisdonk EJ, Kolfschoten N, Verhofstad MH, van der Werken C. Effect of complications after minimally invasive surgical repair of acute achilles tendon ruptures: report on 211 cases. Am J Sports Med. 2011 Apr;39(4):820-4. doi: 10.1177/0363546510392012. Epub 2011 Feb 2. PMID: 21289275.) Therefore, acknowledging high quality studies assessing re-rupture rates after Achilles tondon ruptures is essential in developing recommendations and patient decision aids in the shared decision-making process of treatment selection.

In the Supplementary Appendix to the NEJM-RCT (ref. 28), Supplemental Appendices S2.2: INSTRUCTIONS PROVIDED TO PHYSIOTHERAPISTS RESPONSIBLE FOR THE REHABILITATION OF STUDY PATIENTS. These instructions, which are outlined over three pages, clearly describe the rehabilitation program from injury to 36 weeks post-injury, in detail. The boot was removed 8 weeks after initiation of treatment. The SMART protocol is comparable to the one used in the NEJM-RCT with the exception that the boot is kept for at least 10 weeks in the SMART protocol. This might explain some (but not all) of the re-ruptures in the non-operative group in the NEJM-RCT as half of the re-ruptures in this trial occurred within 10 weeks post-injury. However, all 13 re-ruptures occurred outside a rehabilitation setting and all of them were related to unwanted incidences or poor compliance with excessive loading of the injured foot beyond recommended restrictions, and hence were not the result of inferiority of the post-boot protocol per se.

In the LAMP protocol the initial equinus position is held for 4 weeks and the boot removed at 8 weeks. In the publication of Aujla et al from 2019 almost half of the patients are lost to follow-up and the 9 re-ruptures constitute almost 4% of the 234 patients that were not lost to follow-up (Aujla RS, Patel S, Jones A, Bhatia M. Non-operative functional treatment for acute Achilles tendon ruptures: The Leicester Achilles Management Protocol (LAMP). Injury. 2019 Apr;50(4):995-999. doi: 10.1016/j.injury.2019.03.007. Epub 2019 Mar 11. PMID: 30898390.). Consequently, convincing evidence that the LAMP protocol is superior to other accelerated functional rehabilitation protocols does not exist.

In the Lancet UKSTAR trial from 2020 including 540 adults receiving non-operative treatment with either cast-immobilization or early weight-bearing in a functional brace, no difference in re-rupture rate was observed, implying that early weight bearing and the use of a functional brace do not reduce the risk of re-rupture (Costa ML, Achten J, Marian IR, Dutton SJ, Lamb SE, Ollivere B, Maredza M, Petrou S, Kearney RS; UKSTAR trial collaborators. Plaster cast versus functional brace for non-surgical treatment of Achilles tendon rupture (UKSTAR): a multicentre randomised controlled trial and economic evaluation. Lancet. 2020 Feb 8;395(10222):441-448. doi: 10.1016/S0140-6736(19)32942-3. PMID: 32035553; PMCID: PMC7016510.).

Whether the selection of patients to surgical or non-operative treatment using

ultrasound as in the SMART protocol is associated with lower re-rupture rates is poorly documented. An ongoing RCT will hopefully add new knowledge to the usefulness of using ultrasound as a selection tool (Hansen MS, Vestermark MT, Hölmich P, Kristensen MT, Barfod KW. Individualized treatment for acute Achilles tendon rupture based on the Copenhagen Achilles Rupture Treatment Algorithm (CARTA): a study protocol for a multicenter randomized controlled trial. Trials. 2020 May 12;21(1):399. doi: 10.1186/s13063-020-04332-z. PMID: 32398120; PMCID: PMC7218535.)

Two satellite trials using the CARTA protocol have not shown any superiority of the treatment results using ultrasound as a treatment selection tool (Barfod KW, Overgård AB, Hansen MS, Haddouchi IE, Toft M, Hölmich P. Effect of the Copenhagen Achilles Rupture Treatment Algorithm (CARTA) on Calf Muscle Volume and Tendon Elongation After Acute Achilles Tendon Rupture: A Predefined Secondary Analysis of the First 60 Patients in a Randomized Controlled Trial. Orthop J Sports Med. 2023 Nov 21;11(11):23259671231211282. doi: 10.1177/23259671231211282. PMID: 38021304; PMCID: PMC10664448.) and (Hansen MS, Bencke J, Kristensen MT, Kallemose T, Hölmich P, Barfod KW. Achilles tendon gait dynamics after rupture: A three-armed randomized controlled trial comparing an individualized treatment algorithm vs. operative or non-operative treatment. Foot Ankle Surg. 2023 Feb;29(2):143-150. doi: 10.1016/j.fas.2022.12.006. Epub 2022 Dec 13. PMID: 36528540.). The CARTA protocol uses cast for 3 weeks followed by a functional brace for 6 weeks.

Sample size matters when comparing re-rupture rates between non-operative and surgical treatment of ATR because the re-rupture rates are generally low. The meta-analysis by Ochen et al pooled data from 10 RCTs and 19 observational studies and yielded similar results as in the NEJM-RCT (Ochen Y, Beks RB, van Heijl M, Hietbrink F, Leenen LPH, van der Velde D, Heng M, van der Meijden O, Groenwold RHH, Houwert RM. Operative treatment versus nonoperative treatment of Achilles tendon ruptures: systematic review and meta-analysis. BMJ. 2019 Jan 7;364:k5120. doi: 10.1136/bmj.k5120. PMID: 30617123; PMCID: PMC6322065.)

In conclusion, the length of immobilization in an equinus position as well as time from injury to removal of the boot might be factors affecting the risk of re-rupture and should be investigated further. However, claiming that surgery does not protect against re-rupture does no longer have convincing support in the literature (Barfod KW, Hölmich P. Acute Achilles' Tendon Rupture - Surgery or No Surgery. N Engl J Med. 2022 Apr 14;386(15):1465-1466. doi: 10.1056/NEJMe2202696. PMID: 35417642.).

I will strongly suggest that the authors revise the section in lines 28-37 to adhere to updated strong evidence of increased risk of re-rupture after non-operative treatment.

Please also consider including all the references mentioned above in the reference list.

7. PLOS authors have the option to publish the peer review history of their article (what does this mean?). If published, this will include your full peer review and any attached files.

Reviewer #1: No

Reviewer #2: No

---

## [Author Response · Author response to Decision Letter 1]

15 Apr 2024

Thank you for taking the time to review this submission. Response to reviewers document and revised manuscript attached

---

## [Editor Report · Decision Letter 2]

8 May 2024

Incidence, demographics, characteristics and management of acute Achilles tendon rupture: an epidemiological study

PONE-D-23-38265R2

Dear Dr. Briggs-Price,

We’re pleased to inform you that your manuscript has been judged scientifically suitable for publication and will be formally accepted for publication once it meets all outstanding technical requirements.

Kind regards,

Charlie M. Waugh

Academic Editor

PLOS ONE
---

## [Editor Report · Acceptance letter]

28 May 2024

PONE-D-23-38265R2 

PLOS ONE

Dear Dr. Briggs-Price, 

I'm pleased to inform you that your manuscript has been deemed suitable for publication in PLOS ONE. Congratulations! Your manuscript is now being handed over to our production team.

Kind regards, 

on behalf of

Dr. Charlie M. Waugh 

Academic Editor

PLOS ONE